# Osteoclast-Released Wnt-10b Underlies Cinacalcet Related Bone Improvement in Chronic Kidney Disease

**DOI:** 10.3390/ijms20112800

**Published:** 2019-06-08

**Authors:** Cai-Mei Zheng, Yung-Ho Hsu, Chia-Chao Wu, Chien-Lin Lu, Wen-Chih Liu, Jing-Quan Zheng, Yuh-Feng Lin, Hui-Wen Chiu, Tian-Jong Chang, Jia-Fwu Shyu, Kuo-Cheng Lu

**Affiliations:** 1Graduate Institute of Clinical Medicine, College of Medicine, Taipei Medical University, Taipei 110, Taiwan; 11044@s.tmu.edu.tw (C.-M.Z.); yhhsu@s.tmu.edu.tw (Y.-H.H.); janlin0123@gmail.com (C.-L.L.); wayneliu55@gmail.com (W.-C.L.); jingquan235@gmail.com (J.-Q.Z.); linyf@shh.org.tw (Y.-F.L.); smile710905@gmail.com (H.-W.C.); 2Division of Nephrology, Department of Internal Medicine, School of Medicine, Taipei Medical University, New Taipei City 235, Taiwan; 3Division of Nephrology, Department of Internal Medicine, Shuang Ho Hospital, Taipei Medical University, New Taipei City 235, Taiwan; 4Division of Nephrology, Department of Medicine, Tri-Service General Hospital, National Defense Medical Center, Taipei 10581, Taiwan; wucc@mail.ndmctsgh.edu.tw; 5Division of Nephrology, Department of Medicine, Fu Jen Catholic University Hospital, School of Medicine, Fu Jen Catholic University, New Taipei City 243, Taiwan; 6Division of Nephrology, Department of Internal Medicine, Tungs’ Taichung Metroharbor Hospital, Taichung City 43304, Taiwan; 7Department of Critical Care Medicine, Shuang Ho Hospital, Taipei Medical University, Taipei 235, Taiwan; 8Graduate Institute of Life Sciences, National Defense Medical Center, Taipei 114, Taiwan; 10247@s.tmu.edu.tw; 9Performance Appraisal Section, Secretary Office, Shuang Ho Hospital, Taipei Medical University, New Taipei City 235, Taiwan; 10Department of Biology and Anatomy, National Defense Medical Center, Taipei 110, Taiwan

**Keywords:** cinacalcet, renal osteodystrophy, osteoclast, Wnt 10b, chronic kidney disease

## Abstract

Secondary hyperparathyroidism (SHPT) relates to high turnover bone loss and is responsible for most bone fractures among chronic kidney disease (CKD) patients. Changes in the Wingless/beta-catenin signaling (Wnt/β-catenin) pathway and Wnt inhibitors have been found to play a critical role in CKD related bone loss. A calcimimetic agent, cinacalcet, is widely used for SHPT and found to be similarly effective for parathyroidectomy clinically. A significant decrease in hip fracture rates is noted among US hemodialysis Medicare patients since 2004, which is probably related to the cinacalcet era. In our previous clinical study, it was proven that cinacalcet improved the bone mineral density (BMD) even among severe SHPT patients. In this study, the influence of cinacalcet use on bone mass among CKD mice was determined. Cinacalcet significantly reduced the cortical porosity in femoral bones of treated CKD mice. It also improved the whole-bone structural properties through increased stiffness and maximum load. Cinacalcet increased femoral bone wingless 10b (Wnt10b) expression in CKD mice. In vitro studies revealed that cinacalcet decreased osteoclast bone resorption and increased Wnt 10b release from osteoclasts. Cinacalcet increased bone mineralization when culturing the osteoblasts with cinacalcet treated osteoclast supernatant. In conclusion, cinacalcet increased bone quantity and quality in CKD mice, probably through increased bone mineralization related with osteoclast Wnt 10b secretion.

## 1. Introduction

Secondary hyperparathyroidism (SHPT) is the most common complication of chronic kidney disease (CKD) patients and plays a critical role in renal osteodystrophy (ROD) and cardiovascular disorders [1,2,3]. Constantly elevated parathyroid hormone (PTH) stimulates osteoclast differentiation from circulating hematopoietic progenitors. Increased bone resorption is noted with elevated bone marrow fibrosis and generalized bone loss [4,5,6,7]. Excessive osteoblast activity also follows to compensate for the bone resorption, with resultant osteosclerosis [8]. The degree of mineralization is impaired in such patients because the recently formed bone is removed rapidly without adequate mineralization [9]. Bones formed from such a high turnover status have lower mineralization and trabecular micro-hardness than bones formed from normal or low turnover states [10,11]. Clinically, SHPT patients have been found to have progressive cortical thinning and increased cortical porosity [12,13], which results in decreased cortical bone mineral density (BMD) [14]. Low mineral-to-matrix [10] and carbonate-to-phosphate ratios [15] reduce the bone toughness. Moreover, collagen crosslinking abnormalities have been observed in both serum and soft tissue [16], which were responsible for bone quality loss. Overall, SHPT is associated with both bone quantity and quality loss. 

Since early 2004, a calcimimetic compound, cinacalcet, was widely used clinically among SHPT patients, as it not only inhibited PTH secretion [17], but also arrested the parathyroid gland hyperplasia [18,19]. Other studies also confirmed that cinacalcet can reduce the risk of fractures and improve life quality [20,21]. Our previous study revealed cinacalcet, in combination with vitamin D analogs, improved BMD even among severe SHPT [22]. Cinacalcet has been shown to reverse high bone turnover and bone fibrosis and improve osteoid-related bone parameters among SHPT patients [23,24]. In-vitro studies revealed that activating the calcium-sensing receptor (CaR) in bone cells or high extra-cellular calcium had a bone anabolic effect due to activating osteoblast differentiation [25,26,27] and inducing osteoclast apoptosis [28,29]. However, studies regarding how cinacalcet affects the bone cells and bone mineralization is rare.

Wingless (Wnt) signaling plays an important role in skeletal development and maintenance of bone mass [30,31,32]. Wnts are extracellular proteins that are linked to intracellular Wnt-βcatenin (canonical pathway) [33], noncanonical Wnt–planar cell polarity (WNT-PCP) [34] and Wnt-calcium (WNT-Ca^2+^) [35] pathways when activated. The Wnt ligands, especially Wnt1, Wnt3a, and Wnt10b, activate osteoblast differentiation and increase the rate of bone formation [36,37], whereas Wnt inhibitors sclerostin (SOST) and Dickkopf-1 (DKK1) reduce osteoblast differentiation and survival [38]. Many studies revealed that the Wnt/*β*-catenin pathway plays a definite role in renal osteodystrophy [39,40,41]. In advanced CKD, 15 Wnt genes, including Wnt1, 2, 2b, 3, 4, 5a, 6, 7a, 7b, 8a, 8b, 9a, 9b, 10a and 16 and β-catenin genes, are upregulated [42] with oxidative stress and inflammatory pathways. However, the Wnt signals that improve bone mass, like Wnt10b and Wnt3a, were significantly reduced [42] and Wnt antagonists sclerostin, Dickkopf-1 (DKK1), Secreted Frizzled Related Protein 1 (SFRP1) and SFRP4 levels were significantly increased [41,43] among these patients. Previous studies noted that Wnt10b played an important role in maintaining bone mass by promoting osteoblast differentiation [44], increasing bone mineral density and increasing trabecular numbers [45]. Further, Wnt10b played an important role in osteoclast related osteoblast differentiation and mineralization [46,47]. 

It is proposed that cinacalcet improves both bone quantity and quality in CKD mice through direct effects on bone tissues, at least, by stimulation of osteoclast released cytokines which further improves osteoblast mineralization.

## 2. Results

### 2.1. Cinacalcet Significantly Improved Bone Quantity and Bone Quality in 5/6 Nephrectomy CKD Mice Through Increased Femoral Bone Wnt 10b Expression

The influence of cinacalcet on bone mass was determined in 5/6 nephrectomy CKD mice. Sham and CKD mice without cinacalcet treatment (CKD) were used as controls. The body weight and bone biomarkers, including PTH, Wnt 10b, N-terminal propeptide of type I procollagen (PINP) and C-telopeptide of type I collagen (CTX-1) were measured. CKD without treatment had significantly lower body weight than controls and the CKD+cin group (data not shown). Serum PTH levels were significantly higher in CKD mice than in control mice. CKD+cin had significantly lower PTH levels compared with CKD without treatment after 4 weeks (Figure 1A). Serum Wnt 10b levels were significantly increased in CKD mice and the CKD+cin mice group after 4 weeks (Figure 1B). Serum P1NP levels were significantly decreased in all mice groups after 4 weeks (Figure 1C). However, serum Wnt10b and P1NP levels did not change significantly in CKD+cin mice compared to CKD mice (Figure 1B,C). Serum C-telopeptide of type I collagen (CTX-1) are significantly decreased in the CKD+cin mice group compared with the CKD mice group after 4 weeks (Figure 1D). 

After 4 weeks of observation, the mice were sacrificed and all underwent micro-Computed Tomography (micro-CT) analysis of the femoral bone for bone quantity analysis (Figure 2). Regions of interest containing cortical and trabecular bone were selected for subsequent quantification (Figure 2A). Quantitation of these results indicated that CKD mice had significantly increased cortical porosity and trabecular separation, and reduced cortical bone mineral density, cortical thickness, significantly reduced trabecular volume adjusted for tissue volume and trabecular number (Figure 2B,C). The cortical thickness was non-significantly increased and cortical porosity was significantly decreased in the CKD+cin group (Figure 2B). Trabecular parameters were not significantly affected by cinacalcet treatment in the CKD+cin groups (Figure 2C). The femoral bone structural properties were obtained using a three-point bending test for bone quality analysis (Figure 3A). In CKD mice, the maximum load was significantly decreased compared with controls, whereas, the stiffness and maximum load were significantly increased in the CKD+cin group (Figure 3B). Cortical thickness had a significant negative correlation with total porosity (*p* < 0.001), and also negatively correlated with post-yield displacement (*p* < 0.05). BMD had a significant positive relation with cortical thickness (*p* < 0.05) and a significant negative correlation with post-yield displacement (*p* < 0.01) (Table 1). Bone immunofluorescence staining revealed an increased Wnt10b expression in the femoral bones of CKD+cin compared to the control and CKD mice without treatment (Figure 4).

### 2.2. Cinacalcet Inhibits Osteoclastic Resorption

The in-vitro effects of cinacalcet was demonstrated in primary cell cultures of osteoclast stimulated with 50 ng/mL macrophage colony stimulating factor (M-CSF) and receptor activation of NF-κB ligand (RANKL). The effect of cinacalcet on osteoclast resorptive function was examined by Tartrate-resistant acid phosphatase (TRAP) staining analysis. Under light microscopy analysis, a significant reduction in the bone resorption area was noted with cinacalcet treatment (Figure 5A,B). 

### 2.3. Cinacalcet Increased Osteoclastic Wnt 10b Secretion and Improves Osteoblastic Mineralization

Confocal analysis of immunofluorescent labeling of Wnt10b was greater in osteoclasts treated with 400 nM cinacalcet compared to the control (Figure 5C). Western blot analysis revealed a significant increase in Wnt10b expression among cinacalcet-treated osteoclasts (Figure 5D). Pretreatment with C-59, a Wnt10b secretion inhibitor, further increased the cinacalcet effect of Wnt10b expression within the osteoclasts, and proved that cinacalcet increased Wnt10b released from osteoclasts (Figure 5E). Increased Alizarin Red Stain area was noted as a result of increased mineralization (Figure 5F,G) when calvarial osteoblastic cells were cultured with the supernatant derived from cinacalcet-treated osteoclasts. 

## 3. Discussion

In this study, a novel calcimimetic agent, cinacalcet, was tested for its bone effects in 5/6 nephrectomy CKD mice. Cinacalcet significantly decreased the serum PTH levels after four weeks of therapy compared to other groups (Figure 1A). It increased the bone mass through significant reduction in cortical porosity (Figure 2B) and improved the bone quality through increased maximum load and stiffness (Figure 3B) among CKD animals. These effects probably related with increased Wnt 10b expression in bone cells (Figure 4), which was further proved in in-vitro studies. Cinacalcet inhibits osteoclast bone resorptive function (Figure 5A,B), and increased osteoclast Wnt 10b secretion (Figure 5C–E) which further improved osteoblastic bone mineralization (Figure 5F,G). From this study, it was proved that cinacalcet improved bone mass through direct local effects in the bone micro-environment, at least, by stimulation of osteoclastic Wnt 10b expression. Understanding further molecular mechanisms will be of great clinical significance.

After four weeks of 5/6 nephrectomy, all CKD mice had significantly increased PTH levels compared with sham controls. Previous studies reported that cinacalcet inhibited both parathyroid cell proliferation [48] and hypertrophy [49]. As expected, four weeks of cinacalcet therapy significantly decreased the serum PTH levels in CKD mice. Serum C-telopeptide of type I collagen (CTX-1) which is released during bone degradation was significantly decreased in the cinacalcet treated mice group, and might be explained by reduced osteoclastic bone resorption (Figure 1D). One the other hand, the bone formation marker P1NP level was not significantly decreased, even with lowering PTH levels after cinacalcet treatment, which is possibly related to the preserved osteoblastic bone formation. 

It was revealed that CKD mice had poorer cortical BMD with reduced cortical thickness and significantly increased cortical porosity. Further, the trabecular volume adjusted for tissue volume and trabecular number were also significantly decreased among CKD mice. Both cortical and trabecular bone parameters were disturbed among CKD mice. Nickolas et al. [50] reported that CKD patients had rapid cortical bone loss with significantly lower cortical BMD. Hemodialysis patients revealed a decreased cortical BMD, thinner cortices and increased cortical porosity [51] than age and gender matched control groups. Hyperparathyroidism and increased bone turnover status contributed to cortical bone deterioration in these patients. Cortical bone contributed mostly to whole bone strength since 85% of the peripheral skeleton was composed of cortical bone [52,53]. Previous studies also confirmed that fractures in CKD patients was closely related with significantly decreased cortical bone thickness [51,54,55,56] and cortical bone density [51,54,55]. In this study, it was found that four weeks of cinacalcet treatment increased the cortical thickness and significantly reduced the cortical porosity compared to those without treatment. However, the cortical bone BMD was not significantly increased by cinacalcet, which might be explained by the shorter treatment duration. The trabecular parameters were not significantly affected by cinacalcet. To determine the bone quality, a biomechanical 3-point bending examination was undertaken on the femoral bones of experimental animals. CKD mice exhibited lower maximum load and stiffness than the control ones which proved that they had deteriorated cortical bone structural properties and reduced fracture toughness. Cinacalcet treated mice had a significantly increased maximum load and stiffness, non-significantly decreased post-yield displacement compared with the CKD group, suggesting that cinacalcet improved the cortical bone microstructure and skeletal mechanical integrity. The correlation analysis revealed that cortical bone mineral density had a significant positive correlation with cortical thickness and a significant negative relation post-yield displacement (Table 1). From these findings, this study concluded that cinacalcet treatment improved both bone quantity and quality through improvement in cortical bone parameters. 

Growing evidence indicated that bone formation was closely coupled with bone resorption and that osteoclasts secreted many clastokines [57] which played crucial roles in precise bone formation at previously resorbed sites. Among several clastokines of interest, BMP6 and Wnt10b were found to be released by osteoclasts [47]. The Wnt10b expression in mature osteoclasts stimulated the local differentiation of osteoblasts and enhances osteoblastic mineralization at the end of the resorption phase [47]. Furthermore, Tang et al. [58] revealed that the matrix-bound TGF-β1 actively recruited osteoblast lineage cells to the sites of bone resorption, and served as an effective coupling factor during bone resorption [58]. Ota et al. also revealed that TGFβ1 enhanced Wnt10b expression and secretion from osteoclasts to stimulate osteoblast-directed mineralization [46]. In our recent experiment, it was found that the femoral bones of cinacalcet treated CKD mice revealed an increased in Wnt10b staining compared to CKD mice (Figure 4).

This study further examined the effects of cinacalcet in a bone micro-environment. All bone cells expressed the CaR, and evidence suggested that this receptor played an important role in bone remodeling after a response to extracellular calcium concentration [59]. Herein, it was found that a significant reduction in the bone resorption area after cinacalcet treatment compared to those without treatment (Figure 5A,B). There was unequivocal evidence providing CaR’s role in both osteoclast differentiation and osteoclast function. Cinacalcet might inhibit the osteoclastic bone resorptive function through increased sensitivity to extracellular calcium [60,61]. Further, this study found in the CKD mice that osteoclast numbers were not significantly reduced with cinacalcet (data not shown). Cinacalcet decreased osteoclast bone resorption as determined by TRAP in the in-vitro study, however, it preserved osteoclast number for osteoclast-osteoblast interaction in in-vivo CKD mice.

Further, this study explored the role of cinacalcet in osteoclastic Wnt10b expression and its possible relation to osteoblastic bone mineralization in in-vitro studies. A significantly increased Wnt10b expression was noted in osteoclasts treated with cinacalcet (Figure 5D). In combined treatment with cinacalcet and wnt10b secretory inhibitor, C59, a further increase in intracellular Wnt 10b expression was noted. This proved that cinacalcet treatment increased osteoclast Wnt 10b secretion which was abrogated by C59 (Figure 5E). When the osteoblasts were treated with the supernatant collected from cinacalcet-treated osteoclasts, the osteoblast mineralization was increased as shown by increased alizarin red stain (Figure 5F,G). Thus, this study revealed that cinacalcet increases osteoblastic mineralization, probably through increased osteoclast Wnt10b secretion. To summarize, cinacalcet decreased osteoclastic bone resorption and improved osteoblastic mineralization in bone micro-environment possibly through stimulation osteoclast-osteoblast interaction by increased osteoclastic Wnt 10b secretion (Figure 6).

## 4. Materials and Methods

### 4.1. 5/6 Nephrectomy CKD Model

All animal experiments were performed after approval from the Laboratory Animal Center, National Defense Medical Center (Animal Use Protocol, IACUC-14-027; February, 2014 Taipei, Taiwan). Thirty 6-week to 8-week-old male C57BL/6 mice were purchased from BioLASCO Taiwan Co., Ltd. (Taipei, Taiwan) and acclimated with laboratory conditions at 22 ± 2 °C and 50% ± 10% humidity. Food and water were given ad libitum. The male C57BL/6 mice weighing 18–20 g were anesthetized by intra-peritoneal injection of sodium pentobarbital (50 mg/kg body weight). Then, left nephrectomy was performed through paravertebral cut in the dorsal region. After dissecting of skin, muscle and adipose tissue, the left kidney was exposed. The left renal artery (lower branch) was exposed and ligated with 4-0 silk to induce kidney ischemia. Then, the upper portion of left kidney was removed by cauterization. The right kidney total nephrectomy was done by ligation of renal vessels and renal hilum with 4-0 silk. The whole procedure created 5/6 nephrectomy mice. The retro-peritoneum and muscles were closed by 4-0 nylon suture. The sham mice underwent the same operation through a paravertebral cut as described, the kidneys were exposed, however, no ligation or removal was done. The drug treatment started 4 weeks after 5/6 nephrectomy. The following three groups (6 mice per group) were established: (a) Sham-vehicle (control) group: the sham operation followed by oral gavage treatment with normal saline; (b) CKD-vehicle (CKD) group: CKD followed by oral gavage treatment with normal saline; (c) CKD-cinacalcet (CKD+cin) group: CKD followed by oral gavage treatment with cinacalcet. In the CKD+cin group, mice were administered cinacalcet (10 mg/kg/day, Ohara Pharmaceutical Co., Ltd., Kami Factory, Tokyo, Japan) via oral gavage daily for 4 weeks. Blood samples were obtained using plastic syringes via the retro-orbital vein before and after 4 weeks of drug treatment. The blood samples were allowed to clot at room temperature, after which the serum was separated through centrifugation, collected with aliquots and stored at −80°C before analysis. All the investigators were unaware of group allocation.

### 4.2. Biochemical Analyses

Serum PTH, Wnt10b and Procollagen 1 N-terminal Propeptide (P1NP) levels (*n* = 6 for each group) were measured in all experimental animals by using commercial enzyme-linked immunosorbent assay kits (PTH, Wnt10b and PINP are CUSABIO Biotech Co., Wuhan, China, and CTX-1 are Immunodiagnostic Systems, Boldon, United Kingdom.) according to the manufacturer’s guidance.

### 4.3. Micro-Computed Tomography (μCT)

Mice femur samples were fixed using 4% paraformaldehyde and prepared for further scanning through micro-CT. Bruker Skyscan 1272 (Kontich, Belgium) was used to scan samples at a resolution of 4.6 μm. CT scanning was performed at a voltage of 60 kVp, current of 166 μA, and exposure time of 880 ms with a 0.25-mm aluminum filter. The reconstruction of sections were carried out with GPU-based scanner software (NRecon, Allentown, PA, USA). For trabecular bone analysis in the secondary spongiosa, reconstructed images of the distal femur were isolated. The region of interest was defined as a trabecular bone area of 1.0–3.0 mm below the growth plate (445 slices). In addition, the trabecular and cortical bones were automatically isolated using CTAn software (Version 1.15.4.0, Skyscan, Bruker, Belgium). The morphometric indices or bone mineral density (BMD) of the trabecular bone was also calculated. Bone density reference was validated by BMD calibration phantoms (0.25 and 0.75 g/cm^3^ hydroxyapatite). For 3 D image illustration, CTVox (Version 3.0, Skyscan, Bruker, Belgium) was used.

### 4.4. Biomechanical Three-Point Bending Test

To assess the biomechanical structural characteristics, a three-point bending test was done on the left femora of all animals by a servo hydraulic material testing machine (Bose ElectroForce 3220, Bose Corp, Eden Prairie, MN, USA) (*n* = 6 per group). All frozen bone samples were thawed within a physiological saline solution for 1 h prior to mechanical testing. The entire femoral sample with its physiological curvature pointing up, was attached to a stent with two fixed loading points 8 mm apart. A static 0.5-N preload was applied to the surface of the mid-shaft of the femur located at the midpoint between two lower loading points to stabilize the sample, which was in a perpendicular alignment with the long axis of the femur. A load was applied by controlling movement of the upper loading axis until the final fracture occurs at a constant displacement rate of 0.02 mm/s. The maximum load (N, the maximum tensile load the femur can withstand before fracture), stiffness (N/mm, for elastic deformation) and post-yield displacement (mm, displacement from yield point to yield point) were measured.

### 4.5. Bone Marrow-Derived Monocyte Collection and Osteoclast Differentiation

The osteoclast culture was created as per previous procedures [62]. Bone marrow-derived monocytes were extracted from the tibiae and femurs of 8-week-old male C57BL/6 mice. A total of 1 × 10^6^ cells were cultured in 10-cm culture dishes in α-MEM medium which contained 10% fetal bovine serum (FBS), 50 ng/mL macrophage colony-stimulating factor (M-CSF) and 50 ng/mL RANKL. The medium was changed after 3 days.

### 4.6. Confocal Microscopic Analysis of Osteoclasts

The osteoclasts were cultured on 22 × 22-mm^2^ glass coverslips for 4 days as previously described [63]. The osteoclasts were then treated with 400 nM cinacalcet for 18 h. They were washed with phosphate buffered saline (PBS) and further fixed in 4% paraformaldehyde for 10 min. Then, permeabilization was completed with PBS containing 0.05% Triton X-100 for 1 h. The treated cells were incubated with an antibody specific for Wnt10b (abcam ab66721, USA) in 1% bovine serum albumin in PBS overnight at 4 °C. Cell Navigator™ F-Actin Labeling Kit (Cat No:22663, AAT Bioquest, Inc., Sunnyvale, CA, USA) was used (15 min) to visualize the F-actin distribution. The nuclei were counterstained with Nuclear Red™ DCS1 (1:1000 dilution; AAT Bioquest, Sunnyvale, CA, USA). A confocal microscope equipped with a differential interference contrast optical path (LSM 510, Zeiss, Göttingen, Germany) was used for further imaging. The osteoclasts were considered as successfully cultured if the multi-nucleated cells had more than three nuclei and when more than half of the actin ring was labelled [64]. 

### 4.7. Tartrate-Resistant Acid Phosphatase (TRAP) Staining

The osteoclasts were cultured on 22 × 22-mm^2^ glass coverslips for 4 days and then treated with cinacalcet (400 nM) for 18 h. Next, they were subjected to TRAP staining by a kit containing 50 mM tartrate buffer as per the manufacturer’s instructions (Sigma-Aldrich, St. Louis, MO, USA). At least 500 osteoclasts were counted on three glass coverslips per each treatment. Cells with more than three nuclei were determined as osteoclasts. The number of TRAP positive cells per cover glass was measured using an optical microscope (Axio Imager A2, Zeiss, NY, USA). Briefly, the study set the same red pixel threshold and sum of the total positive area in the control or Cinacalcet treated groups.

### 4.8. Alizarin Red Staining

The osteoclasts were cultured on 22 × 22-mm^2^ glass coverslips for 4 days and then treated with cinacalcet (400 nM) for 18 h. Next, osteoblasts were cultured with the supernatant collected from cinacalcet-treated osteoclasts. Alizarin red stain (ARS) staining was performed using a kit (ScienCell™ 0223, Carlsbad, CA, USA) by slowly adding 1 mL of 2% alizarin red S stain solution to each well. The cells were incubated for 20–30 min at room temperature. The dye was then removed, and the cells were washed 3–5 times with deionized water (diH2O). Subsequently, 1 mL of diH2O was added to each well to prevent cells from drying out. The value of red pixels per coverslip was determined with light microscopy by Axio Imager A2, Zeiss software (Available online: https://www.zeiss.com/microscopy/us/products/light-microscopes/axio-imager-for-polarized-light.html).

### 4.9. Western Blotting

Purified osteoclasts were cultured in the minimum essential medium for 4 days as previously described [63] and then treated with 400 nM cinacalcet for 18 h. The cells were washed with PBS twice and a cold lysis buffer (150 mM NaCl, 50 mM Tris, pH 7.5, 0.25% sodium deoxycholate, 0.1% Nonidet P-40, 1 mM sodium orthovanadate, 1 mM sodium fluoride, 1 mM Pyrolysis of phenylmethylsulfonyl fluoride, 10 mg/mL aprotinin and 10 mg/mL leupeptin) was used to lyse them. The cell lysates were obtained by centrifugation at 16,000 × g for 30 min at 4 °C. Protein concentration was measured by a bicinchoninic acid kit (Pierce, Rockford, IL, USA) and 30 µg of total protein was separated on a 10% sodium dodecyl sulfate polyacrylamide gel. After transferring the protein to a nitrocellulose membrane (Whatman, Dassel, Germany), the membrane was blocked with 5% skim milk in TBS-T (20 mM Tris, pH 7.6, 137 mM NaCl and 0.1% Tween-20) and was incubated with antibodies specific for Wnt10b (abcam ab66721, USA) or actin (Chemicon International, Inc., Billerica, MA, USA). Proteins were visualized through a suitable horseradish peroxidase-conjugated secondary antibody (Santa Cruz Biotechnology, Dallas, TX, USA) and enhanced chemiluminescent reagent (Amersham, Buckinghamshire, UK). Bands were quantified by using densitometry (ProXPRESS Proteomic Imaging System, Perkin Elmer, Melbourne, VIC, Australia) and normalized to loading control actin. The effects of the various treatments were expressed as a fold change relative to the control lanes. Each analysis was repeated for the same procedure for at least three independent experiments.

### 4.10. Statistics

The mean and standard deviation of each value for each group was calculated. Comparisons were made using a post hoc Bonferroni corrected analysis of variance. Data was analyzed using SAS 9.0 (SAS Institute Inc., Cary, NC, USA) and *p* < 0.05 was considered to be statistically significant.

## 5. Conclusions

The study provided the evidence that cinacalcet inhibited the osteoclast bone resorptive function, whereas, increased the osteoclast Wnt 10b release to activate osteoblast mineralization. This study did not explore detail mechanisms on stromal cell populations and this became our study limitation. However, it was definitely proved that cinacalcet improved the bone strength through improving cortical porosity, bone stiffness and maximum load in CKD mice. Future studies regarding cinacalcet effects on osteoclast-osteoblast cross talk in bone microenvironments are still needed.

## Figures and Tables

**Figure 1 ijms-20-02800-f001:**
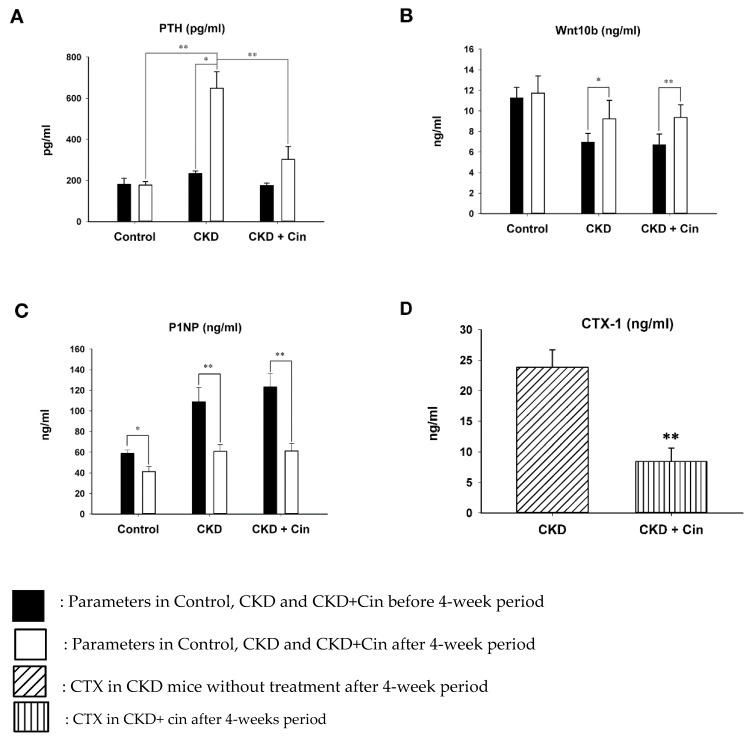
Serum (parathyroid hormone) PTH and Wnt 10b levels before and after 4 weeks in control, non-treated and cinacalcet-treated (chronic kidney disease) CKD mice. (**A**) PTH levels significantly increased in CKD mice compared with controls. The cinacalcet treatment significantly decreased PTH levels in CKD mice. (**B**) Wnt 10b levels increased significantly in both CKD with or without treatment. (**C**) P1NP levels significantly decreased in all groups. (**D**) CTX-1 levels are significantly decreased in CKD+Cin group after 4 weeks treated compared with CKD group. *n* = 6 in each group, * *p* < 0.05, ** *p* < 0.01.

**Figure 2 ijms-20-02800-f002:**
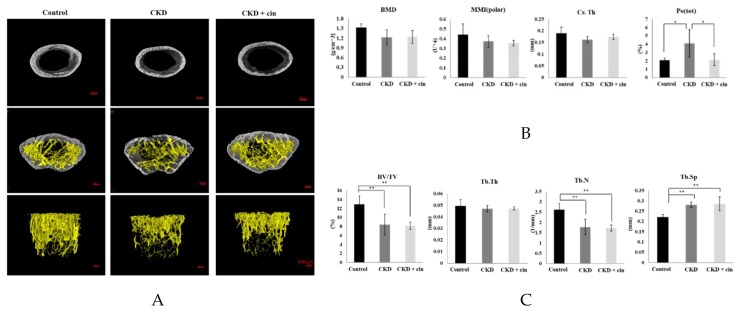
Cinacalcet improves both trabecular and cortical bone microarchitecture. (**A**) Histomorphometric parameters of trabecular and cortical bones of femur in sham-operated (Control) mice, CKD mice treated with saline (CKD), CKD mice treated with cinacalcet 10mg/kg/day (CKD+cin) Scale bar 500 μm. (**B**,**C**) Quantitative results of the experiment shown in A. The cortical porosity significantly improved in cinacalcet treatment group. Cortical bone: Bone mineral density (BMD); cortical thickness (Cr. Th); polar moment of inertia (MMI); total porosity (Po(tot)); bone volume/tissue volume ratio (BV/TV); trabecular thickness (Tb. Th); trabecular number (Tb. N); trabecular separation (Tb. Sp). *n* = 6 in each group, * *p* < 0.05, ** *p* < 0.01.

**Figure 3 ijms-20-02800-f003:**
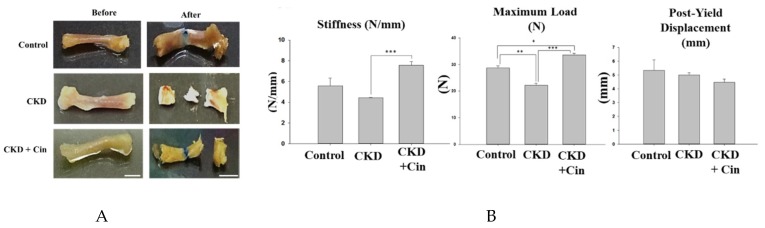
Cinacalcet increases femoral bone structural properties in 5/6 nephrectomy CKD mice. (**A**) Biomechanical three-point bending test of the femoral bone in sham-operated (Control) mice, CKD mice treated with saline (CKD), CKD mice treated with cinacalcet 10 mg/kg/day (CKD+cin) for 4 weeks (left = before; right = after; Scale bar 5 mm). (**B**) Quantitative results of the experiment shown in A. The femoral bone stiffness is significantly increased in the CKD+cin group compared to the CKD group. The maximum load is significantly decreased in the CKD group compared to the control and CKD+cin group. *n* = 6 in each group, * *p* < 0.05, ** *p* < 0.01, *** *p* < 0.001.

**Figure 4 ijms-20-02800-f004:**
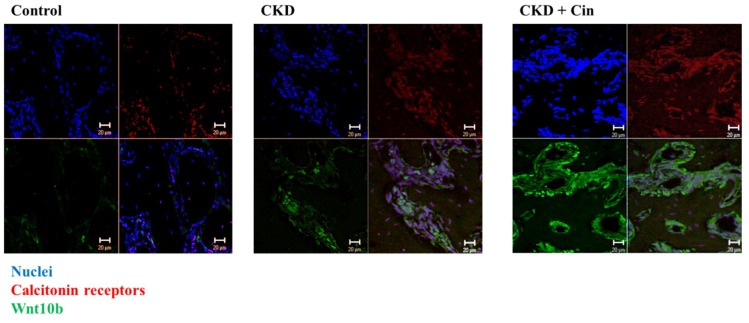
Cinacalcet increases Wnt 10b expression in femoral cortical bones of treated mice. Confocal microscopic analysis of femoral bone cells in the control, CKD, and CKD with cinacalcet treated mice. Blue, nuclei; red, calcitonin receptors; green, Wnt 10b. Scale bar 20 μm.

**Figure 5 ijms-20-02800-f005:**
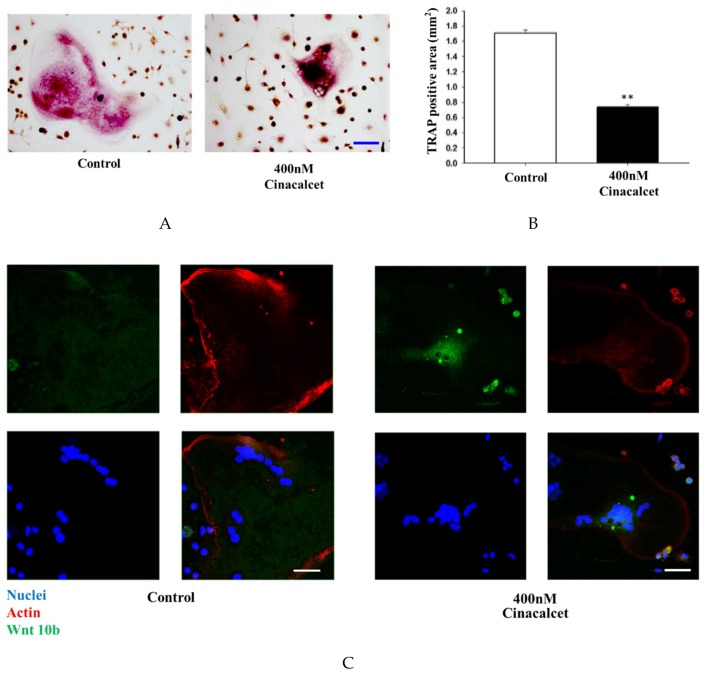
Cinacalcet inhibits osteoclastic bone resorption, increases osteoclast Wnt10b expression and improves mineralization. (**A**) TRAP staining of osteoclasts treated with culture medium alone (control) or culture medium plus Cinacalcet (Cin, 400 nM). Red intracellular staining in the presence of multiple nuclei indicates positive labeling of osteoclasts. Scale bar 20 μm. (**B**) Quantitative results of the experiment shown in A. (**C**) Confocal analysis of osteoclasts treated with culture medium alone (Con) or culture medium plus Cinacalcet (400 nM Cinacalcet). Osteoclasts were labeled with rhodamine-phalloidin (red) to visualize F-actin and TOTO3 (blue) to visualize nuclei. Blue, nuclei; red, actin; green, Wnt10b. Scale bar 20 μm. Each figure represents at least three replicate experiments with a total of at least 500 osteoclasts. (**D**,**E**) Western blot analyses of Wnt 10b expression in osteoclasts. (**D**) Wnt10b expression is significantly increased in osteoclasts treated with Cinacalcet 400 nM. (**E**) Pretreatment with C-59, a Wnt10b secretion inhibitor, further increases the Wnt10b expression by cinacalcet. (**F**) When calvarial osteoblastic cells were cultured with the supernatant derived from cinacalcet-treated osteoclasts, mineralization is increased as expressed by increased Alizarin Red Staining, Scale bar 20 μm, and expressed quantitatively as in (**G**). * *p* < 0.05, ** *p* < 0.01.

**Figure 6 ijms-20-02800-f006:**
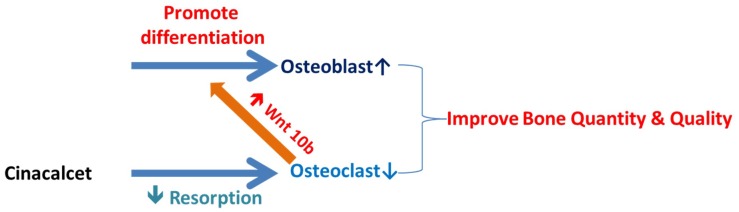
Summary of possible effects of cinacalcet on bones and bone cells. 
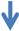
 Decreased; 
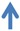
 Increased.

**Table 1 ijms-20-02800-t001:** Correlation between cortical bone parameters.

Cortical Bone Parameters	Total Porosity (%)	Bone Mineral Density (BMD) (g/cm^3^)	Maximum Load (N)	Post-Yield Displacement (mm)	Stiffness (N/mm)
Cortical Thickness	−0.883 ***	0.621 *	0.406	−0.586 *	0.553
Total porosity		−0.500	−0.178	0.380	−0.216
Bone Mineral Denity			0.251	−0.713 **	0.536

* pearson correlation; * *p* < 0.05, ** *p* < 0.01, *** *p* < 0.001.

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
