# Peer review of "Osteoclast-Released Wnt-10b Underlies Cinacalcet Related Bone Improvement in Chronic Kidney Disease"

_ijms, 2019, doi:10.3390/ijms20112800_

Reviewer 1 Report

Zheng et al in the paper entitled "Osteoclast-Released Wnt-10b Underlies Cinacalcet Related Bone Improvement in Chronic Kidney Disease Miceinvestigated in an animal model of CKD the influence of cinacalcet, a calcium-sensing receptor agonist, on bone structure and dissected the effects on osteoclasts. They found that the main player of the benefits on the bone structure was the wnt10b, that is been found upregulated in both the sections of the CKD mice upon cinacalcet treatment and was released by osteoclasts in culture after stimulation with cinacalcet.

Although the topic cinacalcet-bone mass-osteoclast survival is not new, some results are interesting, but some questions arose:

1) In Fig 1, serum levels of PTH and wnt10b were reported. Other markers must be added to complete the description of the cinacalcet effect on bone metabolism and better understand the following results, such as TRAcP, CTX, P1NP, ALP, osteocalcin. Moreover, further control of sham-operated mice + cinacalcet should be added to better uncover possible positive or negative feedback mechanisms due to cinacalcet action on CKD background.

2) In fig 2 and 3, evaluation on sham-operated mice + cinacalcet should be added. In fig 3 author reported a gross evaluation of femurs in two rows. It is supposed to be before (left) and after (right) the test. The author has to better indicate this in the legends and text.

3) Representative pictures of the femur for H&E, TRAcP, ALP/toluidine blue must be showed to better understand the bone phenotype in the experimental groups.

4) In fig 4, immunofluorescences on cortical femur section are reported for calcitonin receptor and wnt10b. Since the main actor in the study is the osteoclast, a colocalization with osteoclast marker has to be provided.

5) In Fig 5 the effect of cinacalcet on osteoclast and osteoblast in vitro cultures. However, some results are a bit hard to understand due to typo/ missing information or mistakes, as below indicated:

6) In 5a, treatment with 400nm cinacalcet is reported. Why is chosen this dosage? how is it linked with the in vivo dose they used before? The author wrote that treatment decreases survival and fusion. While the first conclusion could fit with the representative image but must be documented with the count of TRAcP positive multinucleated cells, the fusion can not be affected by the treatment if they performed as reported in M&M 18h of incubation with cinacalcet after formation of mature osteoclasts.  If this is not the case, please explain better the protocol. Otherwise, incubation starting to time 0 with cinacalcet could reply to this question. However further analysis should be performed other than absolute counts, such as at least the distribution of the number of cells for the number of nuclei i.e 3-8, 9-15, 15-25, more than 25 nuclei). 

7) In 5B, the authors indicated quantitative results for TRAcP staining in fig 5A, but on y-axis wnt10b/actin is reported. It is unclear how this parameter should fit with TRAcP number.

8) In fig 5C, immunofluorescences for blue nuclei and red actin are reported, but a green fluorescent dye is also present. Can the authors clarify what is it?  Why is it absent in control?

9) In fig 5D, the effect on protein expression of wnt10b was evaluated by WB. It is unclear why the author selected actin as the loading control since they show that actin expression and distribution is affected in fig 5C upon cinacalcet treatment. Maybe red ponceau with a different housekeeping protein should be used as loading control.

10) In fig 5E, the effect on osteoblast differentiation upon treatment with osteoclast-conditioned media in different conditions were reported by alizarin red staining. It is unclear why RANKL was used and if RANKL was maintained in cinacalcet treatment. The related graph reported the quantification, but what has been measured? is it surface of positive red cells? If this is the case, do the authors normalized for DNA content or similar?

Minor concerns:

11) Are the described effects meant to be specific for cinacalcet or are common for all the calcium mimetic? maybe a comparison shall be performed for some key experiments between cinacalcet and other members of calcium receptor agonists, such as etelcaltetide.

12) Apoptotic effect on osteoclasts upon cinacalcet treatment is reported in the literature. Do the authors investigate the apoptosis in in vitro or in vivo experiments in their model? 

13) The characters in fig 2 and fig 5 are too small, making very difficult to read the results, please increase the size according to other panels.

14) The cartoon- graphical abstract reported is too complicated and redundant reporting the same images presented in the main body, maybe a simplification will improve the comprehension of the figure.

Author Response

1. Thanks for your great suggestion. We add the results of osteoblastic biomarker, Procollagen 1 N-terminal Propeptide (P1NP) in the discussion (Line 162-164/ Page 7-8) and methods (Line 254/ Page 11). Cinacalcet is a calcimimetic agent used to suppressed high PTH levels in CKD by increasing
calcium sensing receptor sensitivity and suppressing PTH genes within the hyperplastic PTH gland. The sham-operated mice are normal without CKD and normal PTH levels. Thus, we did not perform the control of sham-operated mice + cinacalcet in our study.

2. Thanks for your great advice. We did not perform the control of sham-operated mice + cinacalcet in our study since calcimimetics suppress high PTH levels in CKD and the sham-operated mice are normal without CKD and normal PTH levels. We modify the figures 3 by adding before (left) and after (right) the test, in the legends and text as your comment (Page 24).

3. Thanks for your great suggestion. We did not stain the bone stains including H&E, TRAcP, ALP/toluidine blue, instead we did bone micro-CT scan and bone strength tests to compare the bone changes in between the experimental groups.

4. Thanks a lot for your comment. In this study, we used the calcitonin receptor as osteoclast marker for co-localization and control.
References: calcitonin receptors as osteoclast markers.htm
https://www.ncbi.nlm.nih.gov/pubmed/7664679

5. We make appropriate corrections as your suggestions.

6. Thanks for your correction and we revise the results. In figure 5A, we used 400nm cinacalcet, since wnt 10b increased significantly with this dosage (Figure 5D). We use cinacalcet 10mg/kg/day per oral in our in-vivo 5/6 nephrectomy CKD mice and the detail dose is added in text
(Line 246-247, Page 11).
References:
Cinacalcet HCl attenuates parathyroid hyperplasia in a rat model of secondary hyperparathyroidism. Colloton M1, Shatzen E, Miller G, Stehman-Breen C, Wada M, Lacey D, Martin D. Kidney Int. 2005 Feb;67(2):467-76.

Cinacalcet treatment did not affect the fusion of nuclei during differentiation to osteoclasts; however, it decreases the number of osteoclasts as evident by reduced TRAP-stain positive area. We correct it in our revised manuscript (Line 134,135, page 6)

7. Thanks for your comment. It was labeling error and we correct it in figure 5B.

8. Green fluorescent dye represents wnt 10b staining in the study. It is also present in control, however, fainter than treated group. We improve our image quality in revised manuscript.

9. Thanks for your great comment. In the figure 5C, the actin expression and distribution was not affected by cinacalcet treatment, though polymerization was reduced. Thus, we used actin as traditional housekeeping protein in figure 5D and we replace the figure with clearer one in our revised manuscript.

10. Thanks for your great suggestion. In figure 5F, we cultured the calvarial osteoblastic cells with the supernatant medium derived from cinacalcet-treated osteoclasts. RANKL was used as positive control, since RANKL stimulates osteoclast differentiation. The related graph reported the quantification by measuring the red pixels in the same area. We added similar amounts of osteoblasts to culture media in all experimental groups.

11. Thanks for the great comment. We this this is important and we’ll explore whether the described effects were common for all calcimimetics in our future experiments.

12.  We didn’t investigate the apoptosis pathways in in vitro or in vivo experiments in recent study, however, we have a plan to explore the effects of cinacalcet on apoptosis pathways in our future studies.

13. We modify the figure accordingly.

14. We appreciate the comment and extensively modify the figure as your suggestion.

Reviewer 2 Report

Accept as the format

This paper is trying to elucidate the clinical effect of cinacalcet on bone loss from secondary hyperparathyroidism (SHPT) among chronic kidney disease (CKD) patients. Previous clinical trial has been done to proof the clinical effect of cinacalcet. This is a retroactive study for in vitro and in vivo work using CKD animal model.
This is a relevant study and is a common disease/symphotom found in SHPT patients. Cinacalct is a clinical effective drug and we ought to know the behind mechanism on this.

I don’t think this is not a new topic. Back to 2005, there is clinical evidence showing the effect of Cinacalcet on bone loss (https://www.ncbi.nlm.nih.gov/pubmed/15840675 ). And there is evidence on it may related to Wnt signal (https://www.ncbi.nlm.nih.gov/pmc/articles/PMC5414234/ ). The novelty part for this study is to use a CKD mice model to study, which is different than previous pre-clinical model.

I think the paper is very well written and clear. Easy to follow and read.

The conclusion consist with the evidence and arguments presented, addressing the main question posed.

Author Response

Thanks for your appreciation.

Reviewer 3 Report

The paper from Zheng et al. “Osteoclast-released Wnt-10b underlies Cinacalcet related bone improvement in chronic kidney disease”, reports that Cinacalcet improves bone quality and structure in the CKD mouse model through the induction of Wnt10b release from osteoclasts.

It is an interesting paper with a huge amount of data but the results are often confusedly exposed.

In my opinion, some issues should be address to improve the overall impact and the relevance of this paper. I suggest paying particular attention to all the figures because there are many errors.

Main points

1. Methods: Please provide a detailed description of the 5/6 renal mass reduction model for CKD, and indicate how surgery was performed on sham/control mice. Please, justify the doses of Cinacalcet used in vivo and in vitro and clarify the time of treatment.

2. Figure 1: in Fig. 1A "Δp<0.01" what does it mean? There is not Δ symbol in the histogram. Please delete the zero point in the values in the y axis and provide the axis name (absolute values?).

3. Figure 2: the figure is too small, improve to clarify it. Add the value of the scale bar in the legend of the figure.

4. Table 1 shows the correlation analysis; please provide a better explanation in the text, indicating the type of correlation used.

Pay attention to BMD = bone mineral density. Furthermore, the table should be moved after Figure 3.

5. Figure 3A: Please, specify if the images show the bone before (left) and after (right) the three-point bending test. Again, in the y axis the numbers are too small, please uniform.

6. Figure 4: Please, improve the quality of the figure with higher magnification.

7. Figure 5: increase the font size over the entire figure.

8. -In the histograms (5B and 5G) the authors do not reported the significance, How many replicates have you performed?

9. The histogram in 5B shows the Wnt10b/actin ratio on the y axis. Is it a mistake or what does it mean?

10. -Please specify how the Alizarin Red was performed, I do not understand the 5G histogram that shows the results in mm2.

11. -In 5F, which is the magnification of the control? It looks smaller than the other panels.

12. -In 5C, what is the green channel?

13. -In 5D and 5E, the control columns do not report the standard error. How many independent experiments were performed and how did you evaluate the significance?

14. Figure 6: The Cinacalcet arrow with Wnt 10b must be addressed to the osteoclasts instead of to S1p, BMP6, Wnt 10b. Again, the histograms and the immunofluorescence images are too small and the writing cannot be read (useless with this dimension).

Minor points

15. In the Western blot method, I think the proteins loaded into the SDS-PAGE are 30 μg, not 30 mg.

16. There are many typo and editing errors, please carefully check.

Author Response

1. Thanks for your great suggestion. We describe the procedures for 5/6 renal mass reduction model for CKD, and how surgery was performed on sham/control mice in materials and methods section (Line 232-241/page 10). We use cinacalcet 10mg/kg/day per oral for 4 weeks in our 5/6 nephrectomy CKD mice (Line 246-247, Page 11). References: Kidney Int. 2005 Feb;67(2):467-76. Cinacalcet HCl attenuates parathyroid hyperplasia in a rat model of secondary hyperparathyroidism. Colloton M1, Shatzen E, Miller G, Stehman-Breen C, Wada M, Lacey D, Martin D.

2. Thank you so much for your correction. Δp<0.01 is an error and we remove it from the figure. We also delete the zero point in the values in the y axis and provide the axis name as your suggestion.

3. Thanks for your suggestion. We modify the figure 2 as your suggestion.

4. Thanks for your great advice. We add the explanation of the correlation analysis in the text in results (Line120-128/page 6) and discussion (Line180-184/page 8), move the table to follow Figure 3 (Line613-614/page 25), and correct the typing errors as your suggestion.

5. We appreciate the comment and modify the figure accordingly.

6. Thanks for great suggestion, and we modify the figure with higher magnification.

7. Thanks for your suggestion, and we increase the font size over the entire figure.

8. Thanks for your correction. We add the significance in figures 5B and 5G; we did at least three replicates in each experimental study.

9. Thanks a lot, this was a mistake and we did correction for the histogram 5B erratum.

10. Thanks for the comment. We described how Alizarin red stain (ARS) staining was performed in the material and method section (Line 315-319/Page 13) as follows: Alizarin red stain (ARS) staining was performed using a kit (ScienCell™ 0223) by slowly adding 1 mL of 2% alizarin red S stain solution to each well. The cells were incubated for 20–30 min at room temperature. The dye was then removed, and the cells were washed 3–5 times with deionized water (diH2O). Subsequently, 1 mL of diH2O was added to each well to prevent cells from drying out. The related graph 5G reported the quantification by measuring the red pixels in the same area. We add the Alizarin Red positive area in figure 5G.

11. Appreciate your suggestion. All experiments in figure 5F are under 10x10 magnification, the cell bodies in control are same size as other panels, while, mineralization is less than others.

12. Green channel represents the wnt-10b and we add the label accordingly.

13. In 5D and 5E, we supposed the control as 1, and compare other groups with control. We did at least three replicates in each experimental study, and we evaluate the significance using *p<0.05, **p<0.01. We also add appropriate legends in revised figures (Page 26).

14. This is a great comment, and we do corrections as your suggestions in our revised version.

15. Sorry for typing error and we correct it accordingly (Line 328/Page 14).

16. We did extensive revision and expert editing on the whole manuscript as your suggestion.

Round  2

Reviewer 1 Report

The paper from Zheng et al. entitled "Osteoclast-Released Wnt-10b Underlies Cinacalcet Related Bone Improvement in Chronic Kidney Disease Mice" described the protective role of cinacalcet on the bone loss secondary to chronic kidney disease.
The manuscript is interesting and provides strong evidence supporting their conclusion. However, some points deserve a deeper investigation:
1) In Figure 1, some serum markers are provided, but TRAcP/ NTX levels to understand osteoclast function is mandatory to link the results with the osteoclast activation with the bone improvement.
2) In Figure 4, confocal analysis of cortical part of femurs are provided. However, the co-staining with canonical markers for resident bone cells (TRAcP, Osterix, Osteocalcin, SOST) must be performed to clarify the identity of cells. Moreover, a preliminar histological overview of bones should be provided, such as H&E, TRAcP, toluidine blue staining and correlated cellular parameters (OC n°/BV, OB n°/BV, osteocyte n°/BV) in order to explain the structural parameters shown in Fig 2C/D).
Minor concerns:
3) Scale bars in Fig 3A (both fig and legend) and 5C (fig) must be added.

Author Response

1. Great thanks for your suggestion. Our data showed that significant decrease of serum CTX-1 level in the CKD+Cin group as compared to the CKD group. We add the data in figure 1D and legend, page 3; results section, page 3, line 97-99; and discussion section, page 9, 202-204. In human study, it has been proved that bone biomarkers improved with cinacalcet treatment, including reduction in TRAP and NTX-1 levels with time, however, the effect was through improvement of Ca, P, PTH and mineral metabolism (as reference 1). We found the important local bone effects of cinacalcet as well as systemic mineral regulation in agree with as comment on reference 2.

References

1.      Takashi Shigematsu, Tadao Akizawa, Eiji Uchida, Yusuke Tsukamoto, Manabu Iwasaki, Shouzo Koshikawa, and the KRN1493 Study Group. Long-Term Cinacalcet HCl Treatment Improved Bone Metabolism in Japanese Hemodialysis Patients with Secondary Hyperparathyroidism. Am J Nephrol 2009; 29: 230–236.

2.      Sandro Mazzaferro, Marzia Pasquali. Direct bone effects of calcimimetics in chronic kidney disease? Kidney International 2019, Volume 95 , Issue 5 , 1012 – 1014.

2. Thanks for your questions and great comment. In our preliminary data, we performed the immunohistochemistry analysis of sclerostin expression (brown color) in mice femoral bones as showed below. Decreased sclerostin labeled cells were noted in CKD+Cin group femoral periosteal (green arrow head) and endosteal surface (blue arrow head) and cortical area (red arrow head).                                   

We provide our preliminary histological overview of bone cells using toluidine blue staining and correlated cellular parameters (OC n°/BV). In our data, we found a significant increase in osteoclasts (red arrow head) after cinacalcet treatment in CKD mice. Cinacalcet decreased osteoclast bone resorption as determined by TRAP in in-vitro study, however, preserved osteoclast number for osteoclast-osteoblast interaction in in-vivo CKD mice, we add in our discussion section (page 10, line 250-252).

Minor concerns:
3. We add the scale bars in Fig 3A (both fig and legend) Page 5& 6 and 5C (fig) as in Page 7 as your advice.

Reviewer 2 Report

No comments, Accepted for publication

Author Response

Thanks for your great comment.

Reviewer 3 Report

The paper of Zhen et al. after the revision (second submission) is only slightly improved. I am not completely satisfied by the revision; there are still missing information and errors.

1. Table 1:

When the authors described the table in the text they made errors in the significance reported for cortical thickness vs the post-yield displacement (p<0.1, instead of p <0.05) and for cortical thickness vs BMD (p <0, 1 instead of p <0.05)< span="">

Moreover, when the values are not statistically significant, the authors cannot say that there is a correlation between the parameters considered!

2. Figure 1:

In figure 1 the authors added an histogram in the panel C, however this panel was not described in the result section and in the legend of the figure.

3. Figure 3:

As requested by another reviewer, to better understand the bone phenotype, it would be important to show at least an H&E image for each experimental group.

4. Figure 5:

Fig 5A and 5B:

The authors reported that Cinacalcet reduces TRAP-positive multinucleated osteoclasts. The histogram in fig. 5B reports TRAP positive area, does it mean TRAP-positive cells per mm2? If this is the case, I do not understand the values reported in the Y-axis. Please specify.

5. The 400nM dose of Cinacalcet used in the in vitro experiment how it is related to the dose of 10mg/kg/day used in the in vivo experiment?

6. Fig 5D and 5E:

I agree with the authors when they stated that control was placed as 1. However, it is possible to calculate the standard error on the original data.

7. Fig 5F

Alizarin Red is a dye that selectively binds calcium salts and used to detect and quantify mineralization. Normally after the observation of ARS under an optical microscope, the dye should be desorbed with 10% cetylpyridinium chloride for 1 hour. Then the dye should be collected and the absorbance read at 540 nm in the spectrophotometer.

However, this is not your case, could you please describe your experimental procedure and which kind of software did you use to quantify the red pixels?

Minor points:

8. -In Fig 2A, the indication of the value of the scale bar in the legend of the figure is still missing.

9. -In figure 2C of the histogram for trabecular thickness, in the first lane you indicated "sham Control" uniform to the other histograms (Control)

10. -In the Table 1 there are still errors in the typing of BMD = Bone mineral density not denity!

11. -In Figs 4 and 5C Blue= nuclei (not nuclear or nucleus as reported below the images)

Author Response

1. Thanks a lot and we make appropriate corrections as in results, page 4, line 120-124 and in discussion section, page 10, line 228-232 as your great suggestions.

2. Thanks for your comment. We add the panel in result section, page 3, line 96,97 and the legend of the figure, page 4, 106,107.

3. Thanks for your great comment.

In our preliminary data, we performed the immunohistochemistry analysis of sclerostin expression (brown color) and toluidine blue staining and correlated cellular parameters (OC n°/BV) in mice femoral bones as previously described (as in answer to reviewer 1).

4. Thanks for your great comment. We didn’t explain well in our previous revisions, and we make appropriate corrections in this revised manuscript (In the result section, page 6-7, line 155-158; discussion section, page 10, line 246-248). We calculate the TRAP positive area by the software of AxioVision ® equipped in the microscopy (Image.A2 Carl Zeiss). Briefly, set the same red pixel threshold and sum the total positive area in control or Cinacalcet treated groups. TRAP stain measures the osteoclastic resorptive function, not numbers of osteoclast and the more positive area the more resorptive effects the osteoclast has. We make appropriate corrections in “Materials and Methods”, page 12, line 343-348.

5. Thanks for your great question. Our study is a proof of concept study in order to demonstrate the feasibility of cinacalcet use in CKD related bone disease. We used the 400nM dose of cinacalcet in vitro experiment as usual according with references 1 & 2, whereas, used reference 3 for in-vivo cinacalcet experiment, which is within safety dose for CKD animals.

References

1.      Shalhoub V, Grisanti M, Padagas J, Scully S, Rattan A, Qi M, Varnum B, Vezina C, Lacey D, Martin D. In vitro studies with the calcimimetic, cinacalcet HCl, on normal human adult osteoblastic and osteoclastic cells. Crit Rev Eukaryot Gene Expr. 2003;13(2-4):89-106.

2.      Medina J, Nakagawa Y, Nagasawa M, Fernandez A, Sakaguchi K, Kitaguchi T, Kojima I. Positive Allosteric Modulation of the Calcium-sensing Receptor by Physiological Concentrations of Glucose. J Biol Chem. 2016 Oct 28;291(44):23126-23135.

3.      Cinacalcet HCl attenuates parathyroid hyperplasia in a rat model of secondary hyperparathyroidism. Colloton M1, Shatzen E, Miller G, Stehman-Breen C, Wada M, Lacey D, Martin D. Kidney Int. 2005 Feb;67(2):467-76.

6. Thanks and we totally agree with your comment. In our original data, the values of standard deviations were not significantly different in every situation.
7.  Thanks for your great question.

We performed the Alizarin red stain (ARS) staining using a kit (ScienCell™ 0223) and slowly adding 1 mL of 2% alizarin red S stain solution to each well. The cells were incubated for 20–30 min at room temperature. The dye was then removed, and the cells were washed 3–5 times with deionized water (diH2O). Subsequently, 1 mL of diH2O was added to each well to prevent cells from drying out. The value of red pixels per coverslip was determined with light microscopy by Axio Imager A2, Zeiss software. The method is the same as the calculation of TRAP positive area. We add this in materials and methods section, page 12, line 356,357.

Minor points:

8. Thanks for your comment. We add the value of the scale bar in the legend of the figure, page 5.

9. Thanks for your comment. We replace the figure with corrected one in our revised manuscript as in page 5.

10. Thanks for your comment. We make appropriate corrections in table 1, page 6.

11. Thanks a lot, we make corrections in both figures and legends as in page 6 & 7.

Round  3

Reviewer 1 Report

The Authors consistently replied to the concerns.

Reviewer 3 Report

The authors answered my questions.

However:

- The standard deviation on the control of figures 5D and 5E continues to be missing.

- For the Cinacalcet doses used in vitro and in vivo, please add the references indicated in the answer to my question in the Materials and Methods section.

- In Table 1 there is still Denity instead of Density (not only in the title)